# Identification and Analysis of Aluminum-Activated Malate Transporter Gene Family Reveals Functional Diversification in Orchidaceae and the Expression Patterns of *Dendrobium catenatum* Aluminum-Activated Malate Transporters

**DOI:** 10.3390/ijms25179662

**Published:** 2024-09-06

**Authors:** Fu-Cheng Peng, Meng Yuan, Lin Zhou, Bao-Qiang Zheng, Yan Wang

**Affiliations:** Key Laboratory of Tree Breeding and Cultivation of National Forestry and Grassland Administration, Research Institute of Forestry, Chinese Academy of Forestry, Beijing 100091, China; pfc2121@163.com (F.-C.P.); yuanmeng00829@163.com (M.Y.); zhoulin1214@163.com (L.Z.); zhengbaoqiang@aliyun.com (B.-Q.Z.)

**Keywords:** Orchidaceae, *ALMT* gene family, expression pattern, *Dendrobium catenatum*

## Abstract

Aluminum-activated malate transporter (*ALMT*) genes play an important role in aluminum ion (Al^3+^) tolerance, fruit acidity, and stomatal movement. Although decades of research have been carried out in many plants, there is little knowledge about the roles of *ALMT* in Orchidaceae. In this study, 34 *ALMT* genes were identified in the genomes of four orchid species. Specifically, ten *ALMT* genes were found in *Dendrobium chrysotoxum* and *D. catenatum*, and seven were found in *Apostasia shenzhenica* and *Phalaenopsis equestris*. These *ALMT* genes were further categorized into four clades (clades 1–4) based on phylogenetic relationships. Sequence alignment and conserved motif analysis revealed that most orchid *ALMT* proteins contain conserved regions (TM1, GABA binding motif, and WEP motif). We also discovered a unique motif (19) belonging to clade 1, which can serve as a specifically identified characteristic. Comparison with the gene structure of *AtALMT* genes (*Arabidopsis thaliana*) showed that the gene structure of *ALMT* was conserved across species, but the introns were longer in orchids. The promoters of orchid *ALMT* genes contain many light-responsive and hormone-responsive elements, suggesting that their expression may be regulated by light and phytohormones. Chromosomal localization and collinear analysis of *D. chrysotoxum* indicated that tandem duplication (TD) is the main reason for the difference in the number of *ALMT* genes in these orchids. *D. catenatum* was chosen for the RT-qPCR experiment, and the results showed that the *DcaALMT* gene expression pattern varied in different tissues. The expression of *DcaALMT1-9* was significantly changed after ABA treatment. Combining the circadian CO_2_ uptake rate, titratable total acid, and RT-qPCR data analysis, most *DcaALMT* genes were highly expressed at night and around dawn. The result revealed that *DcaALMT* genes might be involved in photosynthate accumulation. The above study provides more comprehensive information for the *ALMT* gene family in Orchidaceae and a basis for subsequent functional analysis.

## 1. Introduction

Aluminum-activated malate transporters (*ALMT*) are a gene family encoding anion channel protein [1,2,3]. *ALMT* genes share the common feature of containing a conserved structural domain, PF11744, which is comprised of several transmembrane structural domains (TMDs) [4]. *ALMT* genes are widely present in terrestrial plants and encode channel proteins that can transport organic anions (e.g., malate) and inorganic anions (e.g., Cl^−^, SO_4−_, NO^3−^) in cells [5,6]. A large number of studies have demonstrated that the functions of *ALMT* gene family members are involved in plant Al^3+^ tolerance [7,8,9,10], stomatal movement [11,12,13,14,15], mineral nutrition [16,17], and fruit acidity [18,19]. In the C4 and CAM pathways, *ALMT* is mainly involved in the carboxylation phase of malate, which transports malate into the vacuole [20,21]. Furthermore, transmembrane transport of malate across the vacuole membrane has osmoregulatory effects, especially in guard cells, which can drive stomatal movement [22].

The *ALMT* gene family has been identified in several species, and the functions of its single members have been extensively studied, including model plants such as *A. thaliana* and rice [10,13,23] and non-model plants such as apple and soybean [24,25]. Fourteen *AtALMT* genes were identified in *A. thaliana*, which can be divided into four clades. Nine *OsALMT* genes were identified in rice, and clade 5 appeared, containing two members: *OsALMT1* and *OsALMT2*. Subsequent studies demonstrated that its function is related to cellular osmoregulation and the maintenance of electroneutrality [23]. A total of 34 *GmALMT* genes were identified in the soybean genome. In a P-deficient environment, *GmALMT5* improved the utilization of insoluble phosphorus sources and then improved phosphorus efficiency in soybeans [25]. Twenty-five *ALMT* genes were identified in the domesticated apple (*Malus × domestica Borkh.*), and whole-genome duplication (WGD) played an important role in the expansion of the *MdALMT* gene family [24]. In addition, eight candidate *ALMT* genes, including three *ALMT9* homologous genes (*AcoALMT9-1-3*) and five *ALMT1* homologous genes (*AcoALMT1-1-5*), were identified in the CAM plant pineapple (*Ananas comosus*). *AcoALMT1* had higher transcript levels under drought treatment [26]. The functions of single *ALMT* genes have also been studied. The expression levels of *ALMT* genes related to Al^3+^ tolerance (such as *TaALMT1* and *AtALMT1*) were affected by conditions of Al^3+^ and low pH [27]. The overexpression of *StALMT6* and *StALMT10*(*Solanum tuberosum*) in *A. thaliana* enhanced tolerance to Al^3+^ toxicity [28]. *SlALMT4* (*Solanum lycopersicum*) and *SlALMT5* are mainly expressed during fruit development. Their overexpression significantly increases the malic and citric acid content of seeds in the fruit [29]. *AtALMT12* was expressed mainly in leaf guard cells. The almt12 mutant plants have impaired stomatal closure induced by darkness, CO_2_, and ABA, resulting in accelerated plant wilting [14]. *AtALMT6* is also preferentially expressed in leaf guard cells and is involved in stomatal opening [15].

The CAM pathway opens their stomata at night to absorb CO_2_ and close them during the day. This behavior reduces the transpiration loss of water. CAM plants, which have higher water use efficiency (WUE), are better adapted to arid environments compared to the C3 and C4 plants [30,31]. *ALMTs*, as the main bearer of malate transport and stomatal movement, have attracted the interest of many researchers. The diel (diel means diurnal or day/night) expression of the putative *ALMT6* (*Kaladp0062s0038*) gene in the typical CAM plant *Kalanchoë fedtschenkoi* has a distinct circadian rhythm [32]. It is mainly expressed at night and may be related to vacuole malate transport. Two genes homologous to *AtALMT9* (*Aco003023.1* and *Aco010725.1*) are members of the *ALMT* gene family in pineapple [26]. They exhibit the highest expression in leaf photosynthetic tissues and may be prime candidates for the malate influx pathway. *Aco003023.1* is predicted to be a target of miR172d-5p, which is involved in the regulation of photosynthesis (predicted by psRNATarget and TAPIR). The functions of these genes have not been validated in CAM species; therefore, molecular identification and functional characterization of *ALMT* genes in CAM plants are crucial goals.

CAM plants occupy a large proportion of Bromeliaceae, Crassulaceae, Orchidaceae, and Asclepiadaceae [33]. Orchidaceae is the second-largest family of angiosperms, with at least 112 genera showing strong CAM behavior [34]. Orchids are not only prized for their economic and ornamental value but also play a significant role in evolutionary and ecological studies [35]. Whole-genome sequencing of more orchids provides a platform for gene family analyses [36,37,38,39]. Currently, most of the studies on the orchid *ALMT* genes have focused on the transcriptome level, lacking analyses such as structural and evolutionary analyses at the genome level [26,32]. There is a lack of studies that delve into the expression patterns of orchid *ALMT* genes. In this study, we mainly identified the members of the *ALMT* gene family in four orchids, namely *A. shenzhenica*, *P. equestris*, *D. catenatum*, and *D. chrysotoxum*. Next, we analyzed the characteristics, structure, phylogenetic relationship, and cis-acting elements of these members. *D. catenatum* was selected as a representative for expression pattern analysis under the tissue, circadian, and ABA treatments. Our results provide a reference for further clarification of the function of the orchid *ALMT* gene family.

## 2. Results

### 2.1. Identification and Characterization of the ALMT Gene Family in Orchidaceae

Thirty-four *ALMT* genes were identified from four orchid genomes (ten in *D. catenatum* and *D. chrysotoxum*, seven in *A. shenzhenica* and *P. equestris*) (Table 1). The protein sequences of these *ALMT* ranged from 253 to 887 aa. The molecular weights ranged from 27,270.69 to 96,227.92 Da, and the theoretical isoelectric points ranged from 5.7 to 9.39. Seventy percent of the *ALMT* proteins possessed high isoelectric points (pI > 7.0). The majority of them had instability indices below 40 (II), which indicates good stability. The average aliphatic index (AI) of the 34 *ALMT* proteins was 99.87, indicating high thermal stability. In addition, the average hydrophilicity index (GRAVY) of all orchid *ALMT* proteins ranged between 0.5 and −0.5, indicating that they are amphoteric amino acids. Transmembrane structural domains (TMDs) were also predicted, and all *ALMT* gene family members were detected at the N-terminus with numbers ranging from two to seven. Subcellular localization predictions showed that the majority (79.4%) were localized to the plasma membrane. The detailed orchid *ALMT* protein sequences are displayed in Appendix A.

The gene ontology analysis showed significant enrichment of orchid *ALMT* genes in various biological processes (Appendix A). The molecular function was focused on “transmembrane transporter activity”; the most abundant cellular component was the “plant-type vacuole membrane”, and the most abundant bioprocess was the “C4-dicarboxylate transport”. These results suggest that orchid *ALMT* genes are closely related to C4 acid transport and may be involved in dicarboxylate transport in the photosynthetic pathway.

### 2.2. Chromosomal Localization of DchALMT Genes

To reveal the distribution of *ALMT* genes on orchid chromosomes, we mapped the location of *ALMT* using genome annotation information (Figure 1). Since only the *D. chrysotoxum* genome was assembled to the chromosome level, the positions of the *A. shenzhenica*, *P. equestris*, and *D. catenatum* genes on the scaffold were used only for reference (Appendix A). We found that the *DchALMT* gene was unevenly distributed on seven chromosomes. Among them, chromosomes 08 and 16 have two *ALMT* genes at the same locus. We further performed collinearity analysis of *DchALMT* genes using the TBtools MCScanX (version 2.119) tool. The results showed that there were tandem duplications (TD) in two pairs of genes: *DchALMT2/7* and *DchALMT8/10*. Their Ka/Ks values were 0.249092716 and 0.49015544 (Appendix A), suggesting that the *ALMT* genes were subjected to purifying selection during the evolutionary process. Interestingly, even though the *D. catenatum* genome is unassembled to the chromosome, we still found genes located at the same locus: *DcaALMT9/4* and *DcaALMT2/8/5* at the scaffold position. The same situation was not found in *A. shenzhenica* and *P. equestris*.

### 2.3. Multiple Sequence Alignment and Phylogenetic Analysis of ALMT Genes

Previous studies have shown that ALMT-conserved domains contain transmembrane domains (TMD), Gamma-aminobutyric acid (GABA) binding motifs, and WEP motifs [40,41,42]. Multiple sequence alignment of orchid *ALMT* protein revealed that almost all *ALMTs* contain similar conserved sequences (Figure 2). The TMD1 sequence consists of the highly conserved TVVVVVFE sequence, which is preceded by other conserved residues. The transmembrane domain, along with the cytoplasmic helical domain (CHD), constitutes the structural foundation of the ion channel [40]. In previous studies, amino acid residues have been associated with channel activity [43,44]. As a signaling substance, GABA is involved in the regulation of *ALMT*, while *ALMT* can act as a channel for GABA binding and transport [45]. Mutations in residues F213 and F215 can weaken the interactions between the two [46]. We found that some of the F residues are replaced with L in the orchid *ALMT* GABA motif, and the effect of this site on recognition and transport needs to be further investigated. Glutamate E is important for *ALMT* channel activity and was found in *TaALMT1* (E284), *AtALMT1* (E256), and *AtALMT12* (E276) [43,44]. We found that all *ALMTs* contain a conserved WEP motif except *PeALMT7*. Meanwhile, *PeALMT7* lacks multiple conserved residues such as PW, Y, and R. We speculate that *PeALMT7* may have lost channel activity.

To investigate the phylogenetic relationships of the *ALMT* family, phylogenetic trees were constructed using full-length *ALMT* protein sequences from Orchidaceae (34 genes), arabidopsis (13 genes), rice (9 genes), and other species (14 genes) (Figure 3). A total of 70 *ALMT* proteins were classified into four clades, consistent with previous classification in *AtALMT* [3]. Clade 1 had the highest number, and clade 4 had the lowest number of members. There were 10, 7, 8, and 9 orchid *ALMT* members in clades 1–4, respectively. The functions of most genes in clades 1–3 have been characterized, except clade 4.

### 2.4. Gene Structure and Motif Analysis of ALMT Genes

The conserved domains of *ALMT* proteins were searched using NCBI-CD search tools. All orchid *ALMT* genes contain PF11744 structural domains, including some truncated structural domains, such as *DcaALMT5*, *AsALMT4*, *PeAlMT7*, *DcaALMT10*, and *DchALMT4* (Figure 4C). Previous studies have found that truncation of domains leads to altered function [2]. It was also found that *DchALMT7* has a repetitive structure with two almost the same structural domains, like the *ALMT* identified in grape [4].

To understand the structure of orchid *ALMT* genes, 34 *ALMT* proteins were analyzed using the MEME online tool, setting the motif search limit to 20. Most of the *ALMT* proteins had the same conserved motifs in each branch, and the number of *ALMT* motifs ranged from 6 to 29 (Figure 4B, Appendix A). The results showed that the distribution of most *ALMT* motifs is relatively conserved, with motifs 6/15/1/10/9/3/5/17/4/2/8/7 being typical. The TM region corresponds to motif 1, the GABA binding motif corresponds to motif 5, and the WEP motif corresponds to motif 4. In addition, motif19 was found to be present only in clade 1, which may be a unique feature of the orchid *ALMT* clade 1 compared with Arabidopsis (Appendix A). The motif arrangement in clades 2 and 3 was more different, which may predict the functional differentiation in the clade.

Meanwhile, we analyzed the intron-exon structure of the orchid *ALMT* genes (Figure 4D). The results showed that all *ALMTs* contained 4–12 exons and 3–11 introns. We found that compared with *Arabidopsis ALMT*, orchid *ALMT* is a typical long intron type (Appendix A). This has also been found in the *P. equestris* and *D. chrysotoxum* genomes. This has been regarded as a general feature of the orchid genome [37,39]. The longer intron sequences have a higher probability of homologous recombination [47]. This could mean that orchids can produce rich phenotypes in the face of natural selection during evolution.

### 2.5. Cis-Elements in the Promoter Regions of ALMT Genes

To explore the regulatory roles of orchid *ALMTs*, we searched the 2000 bp region upstream of the *ALMT* gene to identify potential cis-acting elements (Figure 5 and Appendix A). We identified a total of 667 cis-acting elements, including 21 types and 14 response functions, in addition to the common TAAT-box and CAAT box (associated with transcription initiation and promotion). The cis-acting elements included responses to phytohormones such as gibberellins, abscisic acid, salicylic acid, auxin, ethylene, and methyl jasmonate; abiotic stresses such as drought, low temperature, defense, and stress responses; and plant growth and development processes such as light response and anaerobic sensing. We found binding sites for *WRKY*, *ERF*, and *MYB* transcription factors, indicating that orchid *ALMT* genes are regulated by multiple transcription factors. Each *ALMT* gene contains multiple elements, of which the most abundant is the light-responsive element (Box 4), followed by ethylene-responsive element (ERE), MYB-binding site, and abscisic acid-responsive element (ABRE). The information on cis-elements is displayed in Appendix A.

### 2.6. Tissue-Specific Expression of DcaALMT

To understand the function of orchid *ALMT* genes, *DcaALMT* genes were selected to determine the expression pattern in four different tissues (roots, stems, leaves, and flowers) by RT-qPCR. The results showed that all *DcaALMT* were expressed in the four tissues except *DcaALMT10* (Figure 6). We found that *DcaALMT1/3/5/6* were highly expressed in leaves, and *DcaALMT2/7/8* were expressed highly in flowers. *DcaALMT4/9/10* were highly expressed in roots. Previous studies have shown that Zea mays *ZmALMT1* is specifically expressed in roots and plays a role in mineral nutrient acquisition and transport [16]. In this study, *DcaALMT4/9/10* were highly expressed in roots, suggesting that they may play a role in root nutrient uptake.

### 2.7. Diel Expression of DcaALMT

To further investigate the expression pattern of *DcaALMTs*, the diel expression patterns of *ALMT* genes in *D. catenatum* leaves were determined and analyzed in combination with the rate of CO_2_ uptake and the content of titratable total acid in the leaves. The results showed that the diurnal changes in CO_2_ uptake rate and titratable total acid in *D. catenatum* were consistent with the characteristics of the CAM pathway (Figure 7A,B). We found that the highest expression of *DcaALMT1/2/3/4/8/9* was at night and dawn, which may be related to CO_2_ uptake and total acid accumulation (Figure 7C). The peak expression of *DcaALMT1/2/4/8* was at night, which was already reduced to a certain level before dawn, and then the expression was lower throughout the photoperiod. The expression of *DcaALMT3/9* was elevated immediately at dawn and reached a peak after 1–2 h (6:00 or 7:00). These results suggest that the expression level of *DcaALMTs* is affected by circadian rhythms and is likely to be involved in the organic acid accumulation of photosynthesis pathway.

### 2.8. Expression of DcaALMT under ABA Treatment

To investigate whether *DcaALMT* is involved in the response to ABA, we used RT-qPCR to study the expression of *DcaALMT* in *D. catenatum* leaves under ABA treatment. The results showed that all *DcaALMT* were sensitive to ABA treatment (Figure 8). Among them, the expression of *DcaALMT3/4/5/8* was up-regulated after ABA treatment, and they reached a peak at 1.5 h or 3 h after treatment and then down-regulated. *DcaALMT5* was the most significantly up-regulated, which was 3.5-fold higher than 0 h. In contrast, *DcaALMT1/2/6/7/9* showed a significant down-regulation after 6 h of treatment, with the most significant down-regulation (0.04) in the expression level of *DcaALMT9*.

## 3. Discussion

*ALMT* was first identified in wheat roots and named after its ability to enhance plant Al^3+^ tolerance [10]. Subsequent studies of *ALMT* over the last decade have revealed that its function is not limited to Al^3+^ resistance but is also associated with stomatal movement, mineral nutrition, fruit acidity, seed development, and GABA signaling [7,11,14,19,23,45,46]. Large-scale plant genome sequencing has laid the foundation for analyzing this family. The *ALMT* gene family has been characterized in the Brassicaceae [42], Poaceae [48,49], Rosaceae [50], and Fabaceae [51,52]. In Orchidaceae, the second largest family among angiosperms, we know little about the *ALMT* family. In this study, 34 *ALMT* genes were identified from four orchid species. The results revealed that each orchid genome contained seven to ten *ALMT* genes, which were fewer than those of *A. thaliana* (14), *Brassica napus* (39), and apple (25). This may be due to the WGD events that occurred in the evolutionary history of Orchidaceae, most of which were accompanied by gene loss after the WGD event [53]. Chromosome distribution showed that 10 *DchALMTs* were distributed on seven chromosomes. *DchALMT10/8* and *DchALMT2/7* were two pairs of tandem duplicated genes. Tandem duplications (TDs) are important for the adaptation of plants to rapidly changing environments [54]. KaKs calculations indicated that these two pairs of genes were subjected to purifying selection during evolution. This is the main reason for the difference in the number of *ALMT* genes between orchids and *A. thaliana*. The same situation occurs in the soybean *MYB* family. *MYB* genes in the soybean show stronger purifying selection, with the appearance of pseudogenes and functionally redundant genes [55]. Purification selection can help plants remove deleterious mutations and retain important genes.

Phylogenetic analyses showed that *ALMT* genes from several species, including Orchidaceae, can be divided into four clades, which is consistent with the *ALMT* family in *A. thaliana* and *Hevea brasiliensis* [56], confirming the reliability of grouping. Phylogenetic, gene structure, and conserved motif analyses of orchid *ALMT* genes revealed that most *ALMT* genes contain conserved N-segment structural domains, which contain 5–7 TMDs (Table 1). The transmembrane region of the N-terminal structural domains is mainly responsible for the transporter activity of the protein [57]. We also found members containing truncated ALMT-conserved domains, such as *AsALMT4*, *DcaALMT5*, *AsALMT2*, and *PeALMT7*. *AtALMT11* represents the same structure that may signify a loss of function within these genes. It is also possible that this is due to incomplete genome annotation, which frequently occurs in the orchid genome [36,39]. In further studies of gene structure, we found that the presence of motif12 only in clade 1 provides more referability to the branching of the orchid *ALMT* phylogeny. The gene structures and motifs of clades 1 and 2 were more conserved within groups, while clades 2 and 3 showed large differences within groups. We speculate that functional divergence may have occurred in clades 2 and 3. In clades 2 and 3, a large number of members (60%) detected novel motifs, mostly concentrated at the C-terminus, with a small concentration at the N-terminus. The C-terminus tends to be associated with independent activity rather than Al^3+^ resistance [58]. Combined with multispecies phylogenetic analyses, *AtALMT4/6/9* in clade 2 and *AtALMT12* in clade 3 were associated with stomatal movement [12,14]. The function of these two clades may be related to the unique stomatal behavior of the CAM pathway. Compared to *Arabidopsis*, the number of introns and exons of the orchid *ALMT* genes is more similar to that of the *AtALMTs*, but the intron length is significantly longer than that of *AtALMTs*. This has become an important feature of orchid species [37]. Longer introns are conducive to the occurrence of homologous recombination, which forms the different combinations of protein domains and enhances phenotypic polymorphism [47].

Gene expression is regulated by upstream transcription factors, making the prediction of cis-acting elements of gene promoters essential. A variety of cis-acting elements were identified in the promoter region of orchid *ALMT* genes, among which light-responsive elements were the most abundant, followed by ethylene, MYB-binding sites, and ABA-responsive elements. These results suggest that light and hormones may affect the expression of the orchid *ALMT* gene. The effects of ethylene on *ALMT* were mainly in Al^3+^ tolerance and fruit acidity regulation [59,60]. In wheat, ethylene can negatively regulate *TaALMT1*, reducing malate efflux from the root and making wheat more sensitive to Al^3+^ toxicity stress [59]. In apples, *MdESE3* can bind to the ethylene-response element (ERE) located in the *MdMa11* promoter and thus activate its expression, promoting malate accumulation [60]. Under drought stress, both *AtALMT4* and *AtALMT12* are regulated by ABA, which in turn closes stomata to reduce water dissipation [61]. *MYB* is an important transcription factor regulating malic acid accumulation. The R2R3-type *MdMYB73* protein directly binds to the promoter of *MdALMT9* to enhance its expression, resulting in the accumulation of malic acid in the vacuole [62]. The above results suggest that Al^3+^ tolerance, stomatal movement, and malate accumulation in the vacuole may be the main direction of the functional study of orchid *ALMT* genes, while the link between *MYB* and orchid *ALMT* needs to be further explored.

Previous studies have shown that *ALMT* has different expression patterns in different plant tissues. For example, *MsALMT1* (*Medicago sativa*) is mainly expressed in roots, which is involved in the response to heavy metal ion stress. *PtaALMTs* (*Pinus tabuliformis*) are specifically expressed in flowers and roots, which may be related to reproductive organs and nutrient uptake. *AtALMT6* and *AtALMT9* are highly expressed in the mesophyll tissue of leaves, and their functions are related to stomatal opening and closing. In our study, the 10 *DcaALMT* genes had different tissue expression patterns. Four of them (*DcaALMT1/3/5/6*) were highly expressed in leaves, three (*DcaALMT2/7/8*) in flowers, and three (*DcaALMT4/9/10*) in roots. In addition, *DcaALMT10* was specifically expressed in roots and flowers. These studies suggest that *DcaALMT* genes may perform different functions in different tissues. The expression of the *BnALMT* genes was up-regulated in roots under phosphorus-deficient conditions. Therefore, we need to further analyze the expression pattern of *DcaALMT* genes to determine the tissue expression pattern under different stress treatments.

An important feature of the CAM pathway is its circadian rhythms [63,64], and the expression of *DcaALMT* genes may also be regulated by the biological clock. We further analyzed the circadian expression pattern of *DcaALMT* genes. The results showed that the expression of *DcaALMT1/2/3/4/8/9* was affected by circadian rhythms and corresponded to diurnal fluctuations in the rate of CO_2_ uptake and titratable total acid. This is consistent with previous reports in the CAM plant *K. fedtschenkoi*. *KfALMT6* and *KfALMT1* are highly expressed at night and dawn [32]. The role of *AtALMT12*, which is located in the same clade as *DcaALMT3*, in stomatal movement has been verified several times [5,14]. *DcaALMT3* was expressed abundantly at dawn, which may lead to states of stomatal closure at a later stage. These states could correspond to the CO_2_ uptake rate curves and total acid curves.

ABA is an important hormone for stomatal closure in response to drought stress in plants [65,66]. Plants regulate the expression of *AtALMT12* through ABA signaling via OST1 and ABI1, leading to stomatal closure [67,68]. Later studies show that xylem sulfate can also directly activate *AtALMT12* expression to accomplish stomatal behavior under drought [69]. Recent studies have shown that *AtALMT4* is also regulated by ABA and mediates stomatal closure by phosphorylation [64]. In this study, the *DcaALMT* genes showed different expression patterns under ABA treatment. *DcaALMT3/4/5/8* reached a peak expression within 1.5–3 h after treatment, whereas *DcaALMT1/2/6/7/9* showed decreased expression after treatment. These results suggest that they may play different functions in the ABA pathway. Stomatal movement is a complex process involving multiple signals and transporters. The Ca^2+^ signal in the guard cell can also regulate stomatal opening and closing [70]. Cytoplasmic GABA signaling can negatively regulate *AtALMT9* to promote stomatal opening [71]. We need to study the expression and function of *ALMT* genes under various signaling molecules to understand how they regulate stomatal movement in CAM plants.

Integration of the CAM pathway into C3 plant crops has been desired as a potential strategy to improve water use efficiency in plants [21]. Previous studies have found that transforming genes from the carboxylation module of the CAM pathway in *Mesembryanthemum crystallinum* into *Arabidopsis* significantly increased plant rosette diameter, leaf area, and leaf fresh weight, and some of the genes also led to an increase in stomatal conductance and titratable total acid accumulation, whereas genes from the decarboxylation module led to a decrease in stomatal conductance and titratable total acid content [20]. Overexpression of the *Populus euphratica* xyloglucan endotransglucosylase/hydrolase gene (*PeXTH*) in *Nicotiana tabacum* decreased intracellular air space in palisade tissues while increasing leaf water content and cellular accumulation, resulting in improved salt tolerance [72]. Our findings that the orchid *ALMT* genes may play a role in stomatal movement and malate transport will help us to better study the stomatal movement module of the CAM pathway. The study of orchid *ALMT* genes in the CAM pathway can help us in the future to accelerate the progress of C3-to-CAM biosystems. This will help to improve the performance of non-CAM plants under stressful environments.

## 4. Materials and Methods

### 4.1. Identification and Bioinformatic Analysis of ALMT Genes in Orchid

Genome data of *A. shenzhenica* [36], *P. equestris* [39], *D. catenatum* [38], and *D. chrysotoxum* [37] were downloaded from the NCBI (https://www.ncbi.nlm.nih.gov/) (accessed on 11 August 2023). The *AtALMT* protein sequence was obtained from the TAIR (https://www.arabidopsis.org/) (accessed on 11 August 2023). The 14 *AtALMTs* protein sequences were used as query sequences, performing a blast on the genomic protein sequence. Next, an online blast was conducted using the NCBI Blastp tool (select the Swissport database). Duplicate sequences were removed after merging the above two results. NCBI CDD search tool and HMMTOP software (version 2.0) were used to identify the conserved domain and transmembrane domains. Combining the results, candidate members that do not contain *ALMT* domains and TMDs were removed. The basic information of *ALMT* protein was predicted using the ExPASY website ComputepI/Mwtool and ProtParam tools (https://web.expasy.org/compute_pi/) (accessed on 11 August 2023). Subcellular prediction is accomplished through Plant mPLo (http://www.csbio.sjtu.edu.cn/bioinf/plant-multi/) (accessed on 11 August 2023). Gene ontology analysis was performed on the eggNOG mapper (http://eggnog-mapper.embl.de/) (accessed on 11 August 2023), and the annotation results were visualized using TBtools [73].

### 4.2. Multiple Sequence Alignment and Evolutionary Tree Construction

Multiple comparisons of orchid *ALMT* protein sequences were performed using the Muscle program in MEGA7 software (version 7.0.14) [74]. The results were visualized and embellished using Jalview software (version 2.10.5) [75], and conserved amino acid sites and structural domains were annotated. The phylogenetic tree was constructed using MEGA7, the method was Maximum Likelihood, and the parameters were set: bootstrap value was set to 1000 times, and the reference model was partial deletion. The evolutionary tree was beautified using Evolview (http://www.evolgenius.info/evolview/) (accessed on 17 August 2023) [76].

### 4.3. Gene Structure and Motif of ALMT Genes

The gene structure (including exons, introns, and UTRs) information was obtained from genome annotation information and visualized using Bio-sequence Structure Illustrator in TBtools [73]. Motif analysis was performed through the online website MEME (http://meme-suite.org/tools/ (accessed on 28 August 2023)) with the following parameters: the number of motifs was set to 20, the site distribution was set to “Zero or one occurrence per sequence”, and the other parameters were kept as default.

### 4.4. Prediction of ALMT Genes Cis-Acting Elements

The 2000 bp upstream of orchid *ALMTs* was extracted by TBtools, and the online site PlantCARE (http://bioinformatics.psb.ugent.be/webtools/plantcare/html/ (accessed on 28 August 2023) was used to identify cis-acting elements in the promoter region [77]. The number of elements and heatmap were visualized using TBtools [73].

### 4.5. Chromosome Localization and Collinearity Analysis

*D. chrysotoxum* with chromosome-level genome was selected as the subject of analysis. The distribution of *ALMT* genes on chromosomes was obtained from genome annotation files; tandem duplicate sequences were searched for using the MCScanX program [78]; visualization was performed using TBtools. The KaKs values of tandem duplicated genes were calculated using the TBtools Simple Kaks calculator [79].

### 4.6. Plant Materials and Growth Conditions

The seed source of *D. catenatum* was introduced from the South China Botanical Garden, Chinese Academy of Sciences, and cultivated in the greenhouse of the Chinese Academy of Forestry.

*D. catenatum* plants were uniformly concentrated in an artificial climate chamber for cultivation and management under photoperiodic dimensions of 12 h darkness and 12 h light, temperature of 26 ± 2 °C, and relative humidity of 60 ± 5%.

The leaves were treated with 100 µmol/L ABA and harvested from the same position after 0, 1.5, 3, and 6 h of the treatment for three biological replicates. The sampling time was from 18:00 to 18:00 the next day. The sampling interval was one hour for 24 h of consecutive sampling and three biological replicates for each sampling site. These samples were placed in liquid nitrogen and stored in a −80 °C refrigerator.

### 4.7. Physiological Index

The Li-6400XT was used to measure the diel CO_2_ exchange rate of *D. catenatum* leaves for 24 h. The instrument was equipped with a 5 L air buffer bottle with a constant gas flow rate (200 μms). Three plants were randomly measured at each sampling point. The readings of one sample point per plant were taken three times with an interval of 30 s.

We conducted the diel titratable total acid of *D. catenatum* leaves for 24 h. Three biological replicates were set up for sampling at each time point, with sample weights of approximately 0.3–0.6 g. The titratable acid content of the leaves was determined by acid-base titration.

### 4.8. Total RNA Extraction and RT-qPCR

RT-qPCR was used to analyze the expression of the *DcaALMT* gene with three biological replicates. The extraction of total RNA was used by the Polysaccharide Polyphenol Plant Total RNA Extraction Kit (TIANGEN; DP441). Easy Script One-Step gDNA Removal and cDNA Synthesis Super Mix (TRANS; AE311) were used for cDNA synthesis. The RT-qPCR was carried out using TB Green Premix Ex Taq (Takara; RR420A) with Roche LightCycler^®^480 Real-Time PCR system. Relative quantification of genes was calculated using 2-ΔΔCT. Primers for RT-qPCR were designed by Primer6 software (version 6.24) (Appendix A) and validated using primer-blast in NCBI.

ANNOVA multiple comparisons were used to analyze the qPCR expression levels between different samples.

## 5. Conclusions

In this study, 34 *ALMT* family members were identified in four orchid genomes and analyzed for their protein basic information and functional annotation. Phylogenetic analyses showed that orchid *ALMT* genes can be divided into four clades. Protein structural domains, conserved motifs, and gene structures analysis showed that clade 1 and clade 4 were conserved across species, while clade 2 and clade 3 were more variable across species, suggesting that functional differentiation may have occurred in clade 2 and clade 3. In *D. chrysotoxum*, tandem duplications were an important driver of *ALMT* evolution and a major cause of differences in gene number between species. Cis-acting element analysis proved that light and hormones are the main factors regulating *ALMT* genes. In addition, the expression patterns of different tissues of *DcaALMT* genes supported the functional differentiation of clade 2 and clade 3. The diel and ABA treatment expression patterns of *DcaALMT* genes predicted a role for *DcaALMT* in the CAM and ABA pathway. In this study, we comprehensively analyzed the basic information and expression patterns of *ALMT* genes in orchids. These results provide insight into the function of *ALMT* genes in the CAM pathway. The next task focuses on the functional validation of *ALMT* genes to improve richer genetic materials for CAM genetic engineering.

## Figures and Tables

**Figure 1 ijms-25-09662-f001:**
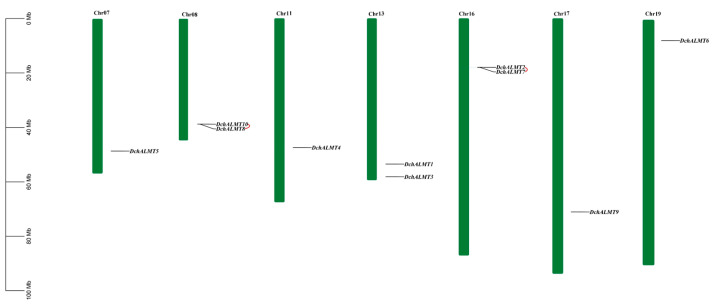
Chromosomal localization of *DchALMT* genes. The red line represents tandem duplicated genes.

**Figure 2 ijms-25-09662-f002:**
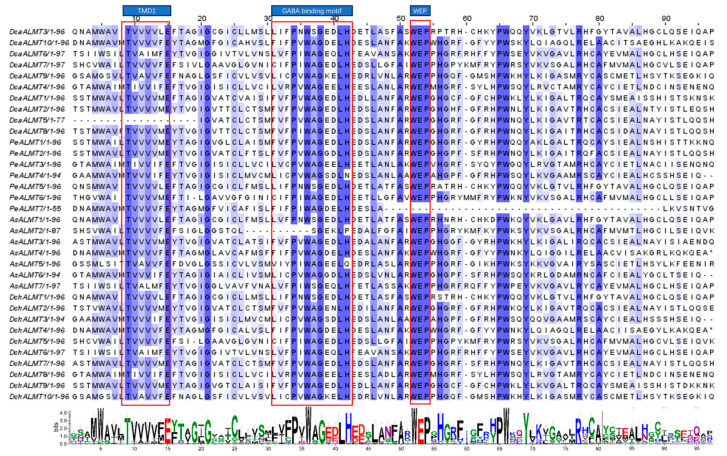
Multiple sequence alignments and typical motifs of orchid *ALMT* protein.

**Figure 3 ijms-25-09662-f003:**
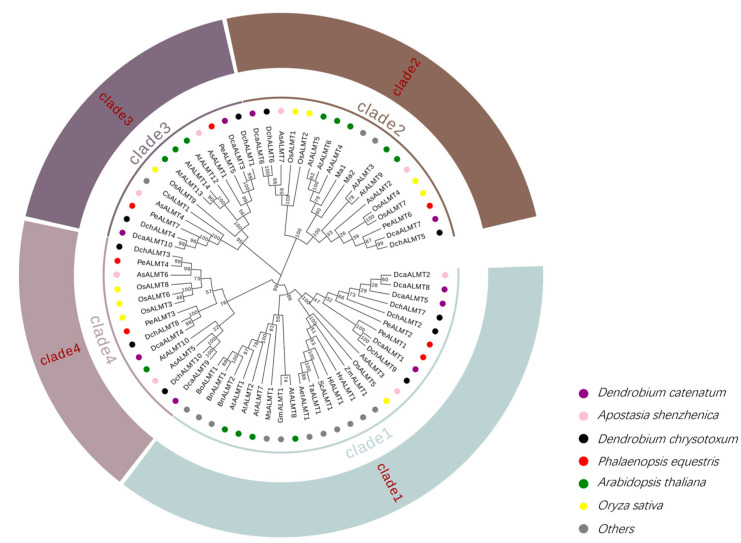
Phylogenetic trees of *ALMT* proteins based on 34 orchid *ALMT* proteins, 13 *AtALMT* proteins, 9 *OsALMT* proteins, and 14 other *ALMT* proteins were constructed by Maximum Likelihood methods. The *ALMT* gene family was classified into four clades. *A. shenzhenica*, *D. catenatum*, *D. chrysotoxum*, *P. equestris*, *A. thaliana*, *Oryza sativa*, *Aegilops tauschii*, *Citrus sinensis*, *Glycine max*, *Holcus lanatus*, *Malus domestica*, *Medicago sativa*, *Triticum aestivum*, *Zea mays*, *Brassica napus*, *Brassica oleracea*, *Hordeum vulgare*, and *Secale cereale* are labeled as Ash, Dca, Dch, Pe, At, Os, Aet, Cs, Gm, Hl, Ma, Ms, Ta, Zm, Bn, Bo, Hv, and Sc, respectively.

**Figure 4 ijms-25-09662-f004:**
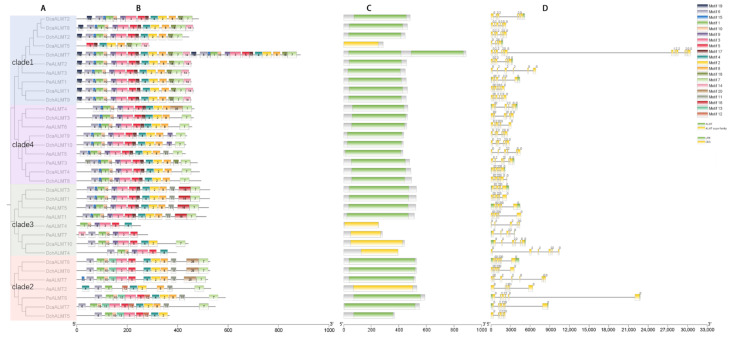
Gene structure, conserved motifs, and domains of *ALMTs*. (**A**) The ML tree contains 34 orchid *ALMTs*. (**B**) Squares of different colors represent conserved motifs of *ALMTs*. (**C**) Squares of different colors represent the conserved domains of *ALMTs*. (**D**) Squares of different colors represent the gene structures.

**Figure 5 ijms-25-09662-f005:**
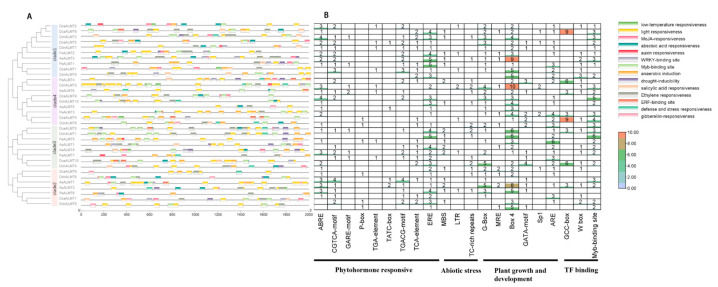
Cis-acting elements in the 2k bp of upstream and downstream regions of orchid *ALMT* genes. (**A**) Elements with similar regulatory functions are displayed in the same color. (**B**) Numbers of each type of element.

**Figure 6 ijms-25-09662-f006:**
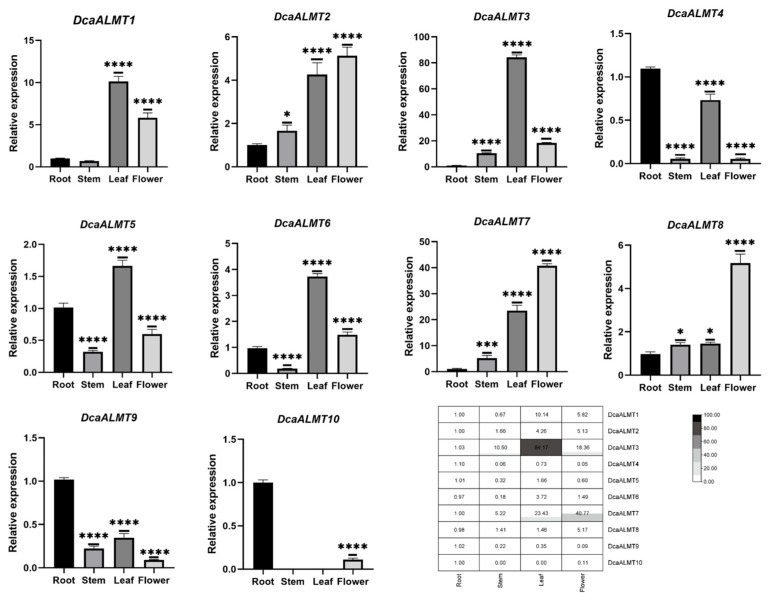
Expression analysis of *DcaALMT* genes at four tissues. The error bars indicate three RT-qPCR biological replicates. Statistical analysis using ANNOVA multiple comparisons for qPCR expression levels between each tissue. The asterisk indicates the P value in the significance test (* *p* < 0.05, *** *p* < 0.001, **** *p* < 0.0001).

**Figure 7 ijms-25-09662-f007:**
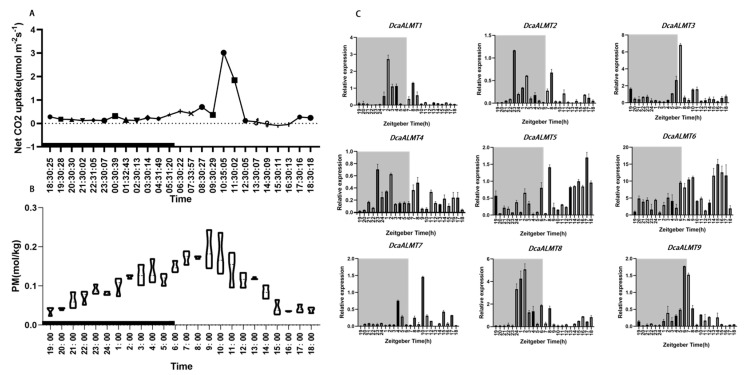
(**A**) The CO_2_ uptake rate of *D. catenatum* leaves over a day and night. (**B**) The fluctuation of titratable total acid over day and night in *D. catenatum* leaves. (**C**) The expression profiles of *DcaALMT* genes over day and night. The black line represents night.

**Figure 8 ijms-25-09662-f008:**
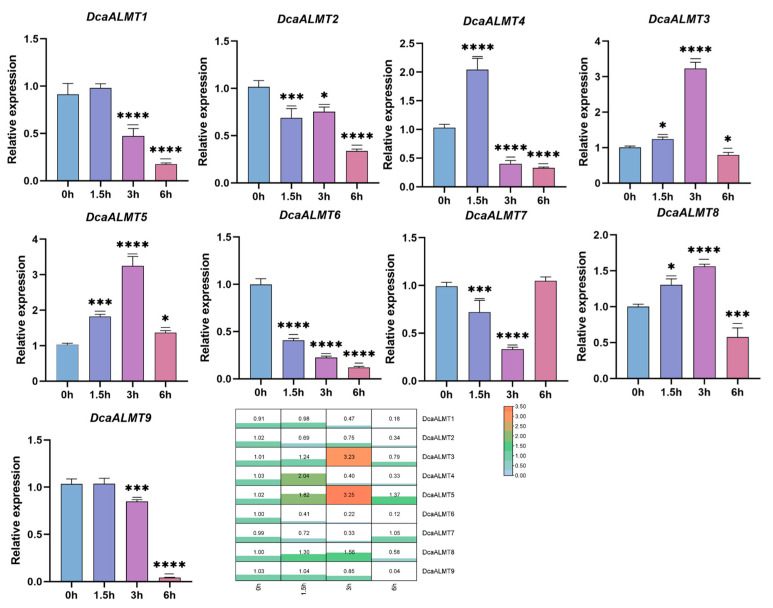
Expression analysis of *DcaALMT* genes at 0 h, 1.5 h, 3 h, and 6 h after ABA treatment. The error bars indicate three RT-qPCR biological replicates. Statistical analysis using ANNOVA multiple comparisons for qPCR expression levels between each time point. The asterisk indicates the P value in the significance test (* *p* < 0.05, *** *p* < 0.001, **** *p* < 0.0001).

**Table 1 ijms-25-09662-t001:** Information on *ALMT* genes in four orchid genomes.

Species	Name	Gene ID	AA(aa)	MW (Da)	pI	II	AI	GRAVY	TMD	Localization
*D. catenatum*	*DcaALMT1*	Dca007047	463	50,348.46	9.02	38.15	99.33	0.179	6	PM
	*DcaALMT2*	Dca006647	482	52,841.68	8.42	37	101.2	0.235	6	PM
	*DcaALMT3*	Dca024716	527	59,656.09	8.85	33.3	96.76	0.04	6	PM
	*DcaALMT4*	Dca000405	486	54,431.84	5.77	40.58	102.94	0.019	6	PM
	*DcaALMT5*	Dca006649	287	49,094.16	8.6	31.2	97.58	0.236	5	PM
	*DcaALMT6*	DcaN03602	527	59,414.26	6.3	36.79	90.34	0.086	5	Vacuole
	*DcaALMT7*	Dca000739	549	61,017.74	6.23	38.94	103.33	0.16	6	ER
	*DcaALMT8*	Dca006648	463	50,712.19	8.9	33.78	99.05	0.19	5	PM
	*DcaALMT9*	Dca000404	435	48,375.72	8.99	32.78	106.92	0.254	7	PM
	*DcaALMT10*	Dca013272	441	48,376.82	8.45	34.65	114.72	0.329	5	PM
*D. chrysotoxum*	*DchALMT1*	Maker66807	527	59,567.04	8.85	35.88	96.56	−0.04	6	PM
	*DchALMT2*	Maker53533	445	48,443.5	8.44	37.5	98.27	0.226	6	PM
	*DchALMT3*	Maker67094	463	50,479.82	6.76	44.6	94.16	0.199	6	PM
	*DchALMT4*	Maker95167	395	42,849.9	9.39	42.92	100.23	0.145	6	PM
	*DchALMT5*	Maker80236	368	40,776.73	8.34	25	103.32	0.375	6	PM
	*DchALMT6*	Maker66167	528	59,246.11	6.26	37.56	89.03	−0.093	7	ER
	*DchALMT7*	Maker53596	887	96,227.92	8.47	33.41	97.63	0.23	12	Extracellular
	*DchALMT8*	Maker58155	493	54,475.68	5.7	39.93	98.56	0.015	5	PM
	*DchALMT9*	Maker39596	454	49,362.17	8.84	40.56	96.58	0.114	6	PM
	*DchALMT10*	Maker58172	433	48,090.46	9.2	32.62	109	0.242	5	PM
*P. equestris*	*PeALMT1*	XM_020717192.1	455	49,782.37	7.63	44.22	93.37	0.072	6	PM
	*PeALMT2*	XM_020720250.1	456	49,218.33	9.13	35.64	100.81	0.238	6	PM
	*PeALMT3*	XM_020726555.1	479	52,949.23	6.12	45.14	98.35	0.04	6	PM
	*PeALMT4*	XM_020728361.1	466	50,732.38	8.23	36.38	96.9	0.231	7	PM
	*PeALMT5*	XM_020727176.1	523	59,234.72	8.9	33.83	95.63	−0.046	6	PM
	*PeALMT6*	XM_020730006.1	590	65,819.3	6.42	36.63	100.15	0.069	6	ER
	*PeALMT7*	XM_020743275.1	282	30,544.02	8.82	26.07	123.81	0.533	6	PM
*A. shenzhenica*	*AsALMT1*	Ash009758	513	58,212.41	8.64	33.54	97.29	−0.083	6	PM
	*AsALMT2*	Ash018925	532	59,665.42	6.37	48.49	93.07	−0.147	4	ER
	*AsALMT3*	Ash011418	447	48,104.86	8.09	41.7	101.79	0.233	6	PM
	*AsALMT4*	Ash007723	253	27,270.69	8.24	37.28	98.73	0.32	5	PM
	*AsALMT5*	Ash020233	431	48,267.47	9.12	37.49	105.23	0.169	7	PM
	*AsALMT6*	Ash011252	457	49,292.44	8.37	33.41	102.98	0.256	6	PM
	*AsALMT7*	Ash005263	520	58,377.16	6.52	40.54	91.93	−0.056	6	ER

## Data Availability

All the data are provided within this manuscript.

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
