# Peer review of "Identification and Analysis of Aluminum-Activated Malate Transporter Gene Family Reveals Functional Diversification in Orchidaceae and the Expression Patterns of Dendrobium catenatum Aluminum-Activated Malate Transporters"

_ijms, 2024, doi:10.3390/ijms25179662_

Round 1

Reviewer 1 Report

Comments and Suggestions for Authors

The manuscript presents an in-depth analysis of the ALMT gene family in four orchid species, focusing on their functional diversification and expression patterns. The study is well-structured and provides significant insights into the role of ALMT genes in Orchidaceae, particularly in Dendrobium catenatum. The identification and categorization of these genes and the phylogenetic and expression analysis contribute to our understanding of ALMT genes in CAM plants. However, several points need to be addressed to strengthen the manuscript.

The study contributes valuable information to the field by extending the knowledge of ALMT genes into the Orchidaceae family, which has been underexplored in this context. However, the authors should emphasize the novelty of their findings more explicitly, particularly in comparison to previous studies in model plants such as Arabidopsis thaliana and rice. The manuscript would benefit from a discussion that clearly articulates what new knowledge or understanding this study brings to plant molecular biology and orchid research.

The results are generally well-presented, but the discussion around the functional implications of the ALMT gene family in CAM photosynthesis and other physiological processes could be more robust. For instance, the manuscript touches on the possible involvement of ALMT genes in stomatal movement and malate transport but does not explore these roles in depth. A more thorough exploration of how the identified genes might contribute to the unique physiological traits of CAM orchids would enhance the impact of the study.

Author Response

Comments 1: The study contributes valuable information to the field by extending the knowledge of ALMT genes into the Orchidaceae family, which has been underexplored in this context. However, the authors should emphasize the novelty of their findings more explicitly, particularly in comparison to previous studies in model plants such as Arabidopsis thaliana and rice. The manuscript would benefit from a discussion that clearly articulates what new knowledge or understanding this study brings to plant molecular biology and orchid research.

Response 1: The authors apologize for confusing the reviewer. In the original manuscript, the context about the number of orchid ALMT family members indicates that WGD and TD are the main driver of evolution. This is the main reason for the difference in the number of ALMT genes between orchids and A. thaliana. Compared to Arabidopsis, the number of introns and exons of the orchid ALMT genes is more similar to that of the AtALMTs, but the intron length is significantly longer than that of AtALMTs. This has become an important feature of orchid species. This helps orchids cope with more complex environments. To give a clearer understanding, in the revised manuscript, the corresponding context has been modified, which can be found in page 10, line 289, 290, 315-317 .

Comments 2: The results are generally well-presented, but the discussion around the functional implications of the ALMT gene family in CAM photosynthesis and other physiological processes could be more robust. For instance, the manuscript touches on the possible involvement of ALMT genes in stomatal movement and malate transport but does not explore these roles in depth. A more thorough exploration of how the identified genes might contribute to the unique physiological traits of CAM orchids would enhance the impact of the study.

Response 2: We think this is a good suggestion. We've added a new discussion on page 12, lines 379-394.

Integration of the CAM pathway into C3 plant crops has been desired as a potential strategy to improve water use efficiency in plants [82]. Previous studies have found that transforming genes from the carboxylation module of the CAM pathway in Mesembryanthemum crystallinum into Arabidopsis significantly increased plant rosette diameter, leaf area, and leaf fresh weight, and some of the genes also led to an increase in stomatal conductance and titratable total acid accumulation; whereas genes from the decarboxylation module led to a decrease in stomatal conductance and titratable total acid content [83]. Overexpression of the Populus euphratica xyloglucan endotransglucosylase/hydrolase gene (PeXTH) in Nicotiana tabacum decreased intracellular air space in palisade tissues while increasing leaf water content and cellular accumulation, resulting in improved salt tolerance [84]. Our findings that the orchid ALMT genes may play a role in stomatal movement and malate transport will help us to better study the stomatal movement module of the CAM pathway. The study of orchid ALMT genes in the CAM pathway can help us in the future to accelerate the progress of C3-to-CAM biosystems. This will help to improve the performance of non-CAM plants under stressful environments.

Reviewer 2 Report

Comments and Suggestions for Authors

Dear Authors,

Reviewer comments ijms-3193036

The manuscript entitled „Identification and analysis of ALMT gene family reveals functional diversification in Orchidaceae and the expression patterns of Dendrobium catenatum ALMT“ represents a useful study aimed at an investigation and characterization of a family of aluminum-activated malate transporters (ALMT) genes and their expression patterns in Dendrobium catenatum following ABA treatment. The study combines bioinformatics with qPCR data gained on Dendrobium catenatum leaves. I can recommend the manuscript for publication in IJMS.

However, I have some comments on the present manuscript which are provided below:

1/ There are only 3 keywords provided in Keywords section. The authors should check the journal´s instructions regarding the minimum and maximum number of keywords.

2/ In Figure 3, appropriate statistical evaluation has to be added to the phylogenetic tree, i.e., numbers at nodes representing bootstrap values per 1,000 replicates, and the algorithm used for the construction of the phylogenetic tree, i.e., maximum likelihood method, has to be given in the figure legend. In Figure 3 legend, line 175, correct the plant name „Brassica oleracea var. … - the variety has to be either specified or removed.

3/ In Figure 6 and Figure 8 legends, the kind of statistical test used for the determination of significant differences has to be specified in the figure legend (e.g., LSD 0.05, Tukey, Duncan multiple range test, etc.) as well as in Materials and methods section.

4/ In Materials and methods, the source of Dendrobium catenatum plants used for the experiments (RT-qPCR) has to be given, i.e., from which institution or natural location the plant material was obtained.

5/ In Materials and methods, the date of access has to be added to all public databases cited, i.e., NCBI, TAIR, ProtParam, Plant mPLo, the eggNOG mapper, and others.

6/ Terminology: I agree that i tis possible to use the term „diel“ for „diurnal“ or „day/night“ cycles; however, I think that the term „diel“ is not so common and should be briefly explained (diel means diurnal or day/night) when used for the first time.

7/ Formal comments on the text related to terminology, English languega nd style:

Materials and methods, part 4.1. heading: Modify the current heading „identification and information of ALMT genes in orchid“ as follows: „4.1. Identification and bioinformatics analysis on ALMT genes in orchid“.

Results, Table 1 heading, line 124: Replace the word „of“ with „on“ in the Table 1 heading: „Information on ALMT genes in four orchid genomes.“

Results, lines 262, 264: Add a space between a number and a corresponding unit in „1.5 h or 3 h“; „6 h“.

Results, line 264: Add „a“ preceding the words „significant down-regulation“ in the statement: „In contrast, DcaALMT1/2/6/7/9 showed a significant down-regulation afetr 6 h of treatment,…“

Discussion, line 271: Replace the word „for“ with „after“ following the verb „named“ in the statement: „ALMT was first idetnified in wheat roots and named after its ability to enhance Al3+ tolerance…“

Final recommendation: Accept after a minor revision.

Comments on the Quality of English Language

Dear Authors,

Reviewer comments ijms-3193036

The manuscript entitled „Identification and analysis of ALMT gene family reveals functional diversification in Orchidaceae and the expression patterns of Dendrobium catenatum ALMT“ represents a useful study aimed at an investigation and characterization of a family of aluminum-activated malate transporters (ALMT) genes and their expression patterns in Dendrobium catenatum following ABA treatment. The study combines bioinformatics with qPCR data gained on Dendrobium catenatum leaves. I can recommend the manuscript for publication in IJMS.

However, I have some comments on the present manuscript which are provided below:

1/ There are only 3 keywords provided in Keywords section. The authors should check the journal´s instructions regarding the minimum and maximum number of keywords.

2/ In Figure 3, appropriate statistical evaluation has to be added to the phylogenetic tree, i.e., numbers at nodes representing bootstrap values per 1,000 replicates, and the algorithm used for the construction of the phylogenetic tree, i.e., maximum likelihood method, has to be given in the figure legend. In Figure 3 legend, line 175, correct the plant name „Brassica oleracea var. … - the variety has to be either specified or removed.

3/ In Figure 6 and Figure 8 legends, the kind of statistical test used for the determination of significant differences has to be specified in the figure legend (e.g., LSD 0.05, Tukey, Duncan multiple range test, etc.) as well as in Materials and methods section.

4/ In Materials and methods, the source of Dendrobium catenatum plants used for the experiments (RT-qPCR) has to be given, i.e., from which institution or natural location the plant material was obtained.

5/ In Materials and methods, the date of access has to be added to all public databases cited, i.e., NCBI, TAIR, ProtParam, Plant mPLo, the eggNOG mapper, and others.

6/ Terminology: I agree that i tis possible to use the term „diel“ for „diurnal“ or „day/night“ cycles; however, I think that the term „diel“ is not so common and should be briefly explained (diel means diurnal or day/night) when used for the first time.

7/ Formal comments on the text related to terminology, English languega nd style:

Materials and methods, part 4.1. heading: Modify the current heading „identification and information of ALMT genes in orchid“ as follows: „4.1. Identification and bioinformatics analysis on ALMT genes in orchid“.

Results, Table 1 heading, line 124: Replace the word „of“ with „on“ in the Table 1 heading: „Information on ALMT genes in four orchid genomes.“

Results, lines 262, 264: Add a space between a number and a corresponding unit in „1.5 h or 3 h“; „6 h“.

Results, line 264: Add „a“ preceding the words „significant down-regulation“ in the statement: „In contrast, DcaALMT1/2/6/7/9 showed a significant down-regulation afetr 6 h of treatment,…“

Discussion, line 271: Replace the word „for“ with „after“ following the verb „named“ in the statement: „ALMT was first idetnified in wheat roots and named after its ability to enhance Al3+ tolerance…“

Final recommendation: Accept after a minor revision.

Author Response

Comments 1: There are only 3 keywords provided in Keywords section. The authors should check the journal´s instructions regarding the minimum and maximum number of keywords.

Response 1: We are very sorry for our careless mistake. Thank you for the reminder. The journal's requirements for the number of keywords are “List three to ten pertinent keywords specific to the article yet reasonably common within the subject discipline.”

We checked the journal's requirement for the number of keywords and revised the keywords to Orchidaceae; ALMT gene family; expression pattern; Dendrobium catenatum in Line 32.

Comments 2: In Figure 3, appropriate statistical evaluation has to be added to the phylogenetic tree, i.e., numbers at nodes representing bootstrap values per 1,000 replicates, and the algorithm used for the construction of the phylogenetic tree, i.e., maximum likelihood method, has to be given in the figure legend. In Figure 3 legend, line 175, correct the plant name „Brassica oleracea var. … - the variety has to be either specified or removed.

Response 2: We think this is a very good suggestion. We have made changes to the figure and the legend.

The legend is modified as follows: Figure 3. Phylogenetic trees of ALMT proteins based on 34 orchid ALMT proteins, 13 AtALMT proteins, 9 OsALMT proteins, and 14 other ALMT proteins were constructed by Maximum Likelihood methods. The ALMT gene family was classified into four clades. A. shenzhenica, D. catenatum, D. chrysotoxum, P. equestris, A. thaliana, Oryza sativa, Aegilops tauschii, Citrus sinensis, Glycine max, Holcus lanatus, Malus domestica, Medicago sativa, Trit-icum aestivum, Zea mays, Brassica napus, Brassica oleracea, Hordeum vulgare, and Secale cereale are labeled as Ash, Dca, Dch, Pe, At, Os, Aet, Cs, Gm, Hl, Ma, Ms, Ta, Zm, Bn, Bo, Hv, and Sc, respectively.

Comments 3: In Figure 6 and Figure 8 legends, the kind of statistical test used for the determination of significant differences has to be specified in the figure legend (e.g., LSD 0.05, Tukey, Duncan multiple range test, etc.) as well as in Materials and methods section.

Response 3: We modified the legends of Figures 6 and 8 and added statistical test types to the material and methods.

Results line 238, Figure 6. Expression analysis of DcaALMT genes at four tissues. The error bars indicate three RT-qPCR biological replicates. Statistical analysis using ANNOVA multiple comparisons for qPCR expression levels between each tissue. The asterisk indicates the P value in the significance test (*p<0.05, ** p < 0.01, *** p < 0.001, ****p<0.0001).

Results line 270, Figure 8. Expression analysis of DcaALMT genes at 0h, 1.5h, 3h, and 6h after ABA treatment. The error bars indicate three RT-qPCR biological replicates. Statistical analysis using ANNOVA multiple comparisons for qPCR expression levels between each time point. The asterisk indicates the P value in the significance test (*p<0.05, ** p < 0.01, *** p < 0.001, ****p<0.0001).

Materials and Methods line 470, 471, ANNOVA multiple comparisons were used to analyze the qPCR expression levels between different samples.

Comments 4: In Materials and methods, the source of Dendrobium catenatum plants used for the experiments (RT-qPCR) has to be given, i.e., from which institution or natural location the plant material was obtained.

Response 4: We list the sources of plants in Materials and Methods 4.6 line 440-442.

The seed source of D. catenatum was introduced from the South China Botanical Garden, Chinese Academy of Sciences, and cultivated in the greenhouse of the Chinese Academy of Forestry.

Comments 5: In Materials and methods, the date of access has to be added to all public databases cited, i.e., NCBI, TAIR, ProtParam, Plant mPLo, the eggNOG mapper, and others.

Response 5: We list the dates of access to public databases in Materials and Methods.

Line 398, NCBI (https://www.ncbi.nlm.nih.gov/ (accessed on 11 August 2023).

Line 400, TAIR (https://www.arabidopsis.org/) (accessed on 11 August 2023).

Line 408, ExPASY website (https://web.expasy.org/compute_pi/ (accessed on 11 August 2023).

Line 409, Plant mPLo (http://www.csbio.sjtu.edu.cn/bioinf/plant-multi/ (accessed on 11 August 2023).

Line 411, eggNOG mapper (http://eggnog-mapper.embl.de/ (accessed on 11 August 2023)

Line 420, Evolview (http://www.evolgenius.info/evolview/ (accessed on 17 August 2023)

Line 425, MEME (http://meme-suite.org/tools/ (accessed on 28 August 2023)

Line 430, PlantCARE (http://bioinformatics.psb.ugent.be/webtools/plantcare/html/ (accessed on 28 August 2023)

Comments 6: Terminology: I agree that i tis possible to use the term „diel“ for „diurnal“ or „day/night“ cycles; however, I think that the term „diel“ is not so common and should be briefly explained (diel means diurnal or day/night) when used for the first time.

Response 6: We explained the first occurrence of “diel” in the text.

Line 76, The diel (diel means diurnal or day/night) expression of the putative ALMT6 (Kaladp0062s0038) gene in the typical CAM plant Kalanchoë fedtschenkoi has a distinct circadian rhythm [32].
